# OpenReview forum: "Efficient Large Language Models Moderation with Multi-Layer Latent Prototypes"
_ICLR.cc/2026/Conference — ICLR 2026 Conference Desk Rejected Submission_

### Official Review · Reviewer_86ZE · 2025-10-16

**Soundness:** 3
**Presentation:** 3
**Contribution:** 2
**Rating:** 4
**Confidence:** 4

**Summary:**

The paper focuses on the problem of LLM input moderation where if a user queries an LLM with a harmful prompt, the LLM should be able to detect it and abstain from generating an output. The paper proposes LPM, a method that uses last token hidden representations from multiple layers of the LLM to compute class (sub-class) conditional prototypes and trains a set of aggregated weights to add sparsity.  At inference, LPM computes a safety score for a test prompt from select hidden layers and combines them via learned weights to produce P("\"unsafe\""). Experiments on various harmful and neutral datasets show both the effectiveness and efficiency of LPM.

**Strengths:**

- The paper is generally easy to follow, though there is room for improvement as discussed below.

- The experiments are quite comprehensive, covering multiple datasets and models, and also include thorough ablation studies.

- The method is quite efficient and does not require full-scale training/fine-tuning or dataset curation, unlike guard models.

**Weaknesses:**

- I have some reservations about the novelty of the work. Prior studies have already explored the use of LLM embeddings for input moderation (e.g., Abdelnabi et al., 2025; Ayub & Majumdar, 2024), and the paper does not clearly articulate how LPM meaningfully differs from these approaches. While it notes that “previous methods that utilize latent representations do not incorporate the knowledge from multiple layers, …”, if this is the primary distinction (and/or the use of Mahalanobis distance instead of Euclidean distance), the contribution may come across as somewhat incremental.

- I think the paper could be strengthened by adding more competitive guard models e.g. Granite Guardians [1], ShieldGemma [2], SafePhi [3] which have shown to outperform baselines used in this paper (e.g. WildGuard, LlamaGuard).

- I noticed the main results don’t show error bars or standard deviations (e.g. Tables 1-4, Figures 2-3). It’d be great to include them since they help show how consistent and reliable the results are, especially with randomness in training or data sampling. Error bars help assess the statistical significance and reproducibility of findings, which are critical for drawing reliable scientific conclusions.

(Minor):
- The equations could benefit from clearer explanations. For example, the notation is somewhat confusing: in lines 147–151, \(\mu_c\) is defined as the average of class in the *sample space*, whereas in Equation (4), \(\mu\) refers to the average of *layer-wise representations*.
-The notation in Equation (4) could be improved for clarity. For instance, use \(h_l\), \(\mu_l\) instead of \(h_i\), \(\mu_i\) to indicate the layer index.


- In the Conclusions section, there is redundancy in the first two sentences (lines 472–477), as they convey nearly the same message. Also, on line 482, there is a missing full stop in the phrase: “moderation needs LPM represents.”

[1] Padhi, I., Nagireddy, M., Cornacchia, G., Chaudhury, S., Pedapati, T., Dognin, P., Murugesan, K., Miehling, E., Cooper, M. S., Fraser, K., et al. Granite guardian. arXiv preprint arXiv:2412.07724, 2024.

[2] Zeng, W., Liu, Y., Mullins, R., Peran, L., Fernandez, J., Harkous, H., Narasimhan, K., Proud, D., Kumar, P., Radharapu, B., et al. Shieldgemma: Generative ai content moderation based on gemma. arXiv preprint arXiv:2407.21772, 2024a.

 [3] N. Machlovi, M. Saleki, I. Ababio, and R. Amin, “Towards safer ai moderation: Evaluating llm moderators through a unified benchmark dataset and advocating a human-first approach,” arXiv preprint arXiv:2508.07063, 2025.

**Questions:**

-	Why does LPM use the last token embedding? What is the rationale behind this decision? do we have ablation on other tokens?

-	How is LPM different from methods that also use embeddings of the models (Abdelnabi et al., 2025; Ayub & Majumdar, 2024)? Is the difference only the use of multiple players? And/or the use of Mahalanobis distance instead of Euclidean distance ?

---

> ### Author Response · Authors · 2025-11-19
> **Response (1/2)**
>
> We appreciate the reviewer’s feedback and address the mentioned points below.
>
> > Why does LPM use the last token embedding? What is the rationale behind this decision? do we have ablation on other tokens?
>
> The design of our method is driven by simplicity and effectiveness. We aim to make our method as performant as possible while retaining the best safety assessment capabilities. The use of the last token is the most cost-effective solution, which is also applicable to all models and performs well in practice.
>
> In autoregressive models, earlier token representations do not attend to future tokens, so any incident of harmful content can only be encoded in the representations of the tokens when it occurs and in the representations of future tokens. Since we do not know when the possibly harmful tokens will start to appear, it is impossible to determine a priori which subset of tokens we should aggregate for safety assessment most efficiently. However, in instruction models that we mainly target, the last token will always have to encode the information about ‘what content to generate in the future given the tokens seen in the past’, so it’s the most natural and robust choice. Current work[1] highlights that the models encode information about what they will generate, which additionally strengthens our choice of last token representation, as the model will encode unsafe intent in the representation of the last token of the prompt.
>
> Using any other single token would require additional methods to choose, which is not trivial given the variable input length and no prior knowledge about the content of the prompt. The use of more than one token would require aggregation techniques, which would introduce complexity and additional computation to our method. In the Table below, we compare LPM performance when using the last token and the mean of all prompt tokens.
>
> | **Representation** | **Llama-8B-Inst** | **Mistral-7B-Inst** | **OLMo2-7B-Inst** |**Qwen3-8B-Inst**|
> |-|-|-|-|-|
> | Last token | **88.30** | **87.55** | **90.18** | **85.95** |
> | Mean over all tokens | 86.59 | 84.22 | 88.32 | 83.46 |
>
> **This further empirically supports our choice of last token representations for LPM.** We updated the paper to include these results in Appendix H  and reference them in the main paper.
>
> > I have some reservations about the novelty of the work. Prior studies have already explored the use of LLM embeddings for input moderation (e.g., Abdelnabi et al., 2025; Ayub & Majumdar, 2024), and the paper does not clearly articulate how LPM meaningfully differs from these approaches. While it notes that “previous methods that utilize latent representations do not incorporate the knowledge from multiple layers, …”, if this is the primary distinction (and/or the use of Mahalanobis distance instead of Euclidean distance), the contribution may come across as somewhat incremental.
>
> To our knowledge, **LPM is the first method to use learned prototypes for input safety assessment**, which we show yields consistent, significant gains over other classification approaches as we demonstrate in. Table 1 and Appendices F and G.
>
> Moreover, we also **leverage multi-layer representations for safety assessment, which we view as a non-trivial contribution**. Again, as far as we know, LPM is the first application of multi-layer sparse feature selection to input moderation.
>
> We highlight that the results reported in Table 1 for Abdelnabi et al., 2025 and Ayub & Majumdar, 2024 reflect their best-case performance using OLMo embeddings. The performance of these methods drops substantially when applied to embeddings from other models. We have updated Table 1 to include detailed results across multiple model embeddings, **which clearly show that LPM outperforms the alternatives, particularly for non-OLMo embeddings.**
>
> While our method is simple, we do not view it as incremental, and the simplicity of LPM is a practical strength that makes it easy to adopt. We offer extensive evaluations and ablations of both our approach and prior methods, which constitute additional contributions of our work. LPM consistently outperforms alternative existing techniques. Taken together, these points highlight how LPM represents a meaningful, practically relevant, and non-incremental contribution.
>
> > I think the paper could be strengthened by adding more competitive guard models e.g. Granite Guardians [1], ShieldGemma [2], SafePhi [3] which have shown to outperform baselines used in this paper (e.g. WildGuard, LlamaGuard).
>
> We thank the reviewer for suggesting these models. We have added them as additional baselines to Table 1, excluding Safe-Phi, as we were unable to achieve scores higher than F1 > 0.3 following guidelines on HuggingFace. **LPM outperforms all of those models under the presented evaluation protocol.**

---

> ### Author Response · Authors · 2025-11-19
> **Response (2/2)**
>
> > I noticed the main results don’t show error bars or standard deviations (e.g. Tables 1-4, Figures 2-3). It’d be great to include them since they help show how consistent and reliable the results are, especially with randomness in training or data sampling. Error bars help assess the statistical significance and reproducibility of findings, which are critical for drawing reliable scientific conclusions.
>
> For the guard models, we use greedy decoding, which yields deterministic results.. For other methods, we report the average over 5 random seeds. In instances where a larger variance in results occurred, we included confidence intervals (e.g., Figure 5(b)). However, the standard deviation in our main tables for LPM is very small (<0.1), so we omitted them to avoid obscuring the presentation of our results.
>
> For the reviewers' sake, we provide standard deviations for our main results in Table 1 in below
>
> | **Model** | **Standard Deviation of F1** |
> |:-|-:|
> | LPM(Llama-8B-Inst) | 0.0401147 |
> | LPM(Mistral-7B-Inst) | 0.0214627 |
> | LPM(OLMo2-7B-Inst) | 0.0631332 |
> | LPM(Qwen3-8B-Inst) | 0.0440776 |
>
> We have updated the introduction to the Experiments section to highlight that our results are calculated as an average over 5 random seeds. If the reviewer considers standard deviations to be of high importance, we are open to incorporating them into the final version of the article.
>
> > The equations could benefit from clearer explanations. For example, the notation is somewhat confusing: in lines 147–151, (\mu_c) is defined as the average of class in the sample space, whereas in Equation (4), (\mu) refers to the average of layer-wise representations.-The notation in Equation (4) could be improved for clarity. For instance, use (h_l), (\mu_l) instead of (h_i), (\mu_i) to indicate the layer index.
>
> > In the Conclusions section, there is redundancy in the first two sentences (lines 472–477), as they convey nearly the same message. Also, on line 482, there is a missing full stop in the phrase: “moderation needs LPM represents.”
>
> We thank the reviewer for bringing this to our attention. We have updated our work accordingly
>
> We hope our responses address the reviewer’s concerns and make them reconsider a positive evaluation of our work. If the reviewer has any further concerns or additional points to raise, we are eager to address them.
>
> **References**
>
> [1] Lindsey, et al., "On the Biology of a Large Language Model", Transformer Circuits, 2025.

---

### Official Review · Reviewer_5bZL · 2025-10-18

**Soundness:** 4
**Presentation:** 3
**Contribution:** 2
**Rating:** 4
**Confidence:** 3

**Summary:**

This paper proposes a lightweight LLM moderation framework, *Latent Prototype Moderator* (LPM), designed to assess input safety and prevent LLMs from generating potentially harmful content. LPM constructs *safe* and *harmful* prototypes using hidden representations from multiple LLM layers, enabling classification via Gaussian Discriminant Analysis (GDA) for new inputs. The proposed approach achieves state-of-the-art performance on several input moderation benchmarks and can be combined with output steering methods for improved moderation performance.

**Strengths:**

1. The proposed LPM is conceptually intuitive and presented in a clear, accessible manner.
2. Experimental results demonstrate that LPM performs competitively across benchmarks, achieving strong performance comparable to more computationally intensive baselines that require additional training.
3. The authors provide several analyses that help clarify the behavior and effectiveness of LPM.

**Weaknesses:**

1. My primary concern is that the proposed LPM reuses several techniques that have previously been applied to LLM trustworthiness and safety-related tasks (e.g., [1]). The main innovation appears to be aggregating classification results across multiple layers, using a method akin to LASSO for feature selection. While the empirical results are solid, the conceptual contribution seems incremental, placing the paper on the borderline of acceptance.
2. Given the simplicity of the proposed method, the paper could be strengthened by providing deeper insights into the underlying mechanisms that make LPM effective—particularly regarding what properties of the LLM’s hidden states enable accurate moderation.
3. The evaluation primarily focuses on input classification metrics such as F1 score. However, input moderation alone may not capture the nuanced trade-offs in real-world safety scenarios. For instance, OpenAI’s *safe completions* approach [2] demonstrates that maintaining safety while providing helpful responses (e.g., offering emotional support in self-harm cases) may require output-centric safety mechanisms rather than pure input filtering. Hence, the evaluation could be more convincing if it discussed the broader context and limitations of input-only moderation.

> **References**
>
> [1] *Representation Engineering: A Top-Down Approach to AI Transparency.*
>
> [2] *From Hard Refusals to Safe Completions: Toward Output-Centric Safety Training.*

**Questions:**

1. How does LPM handle *dual-use* scenarios [2], where user queries may have both benign and harmful interpretations (e.g., asking for the minimum energy to ignite a firework display)?
2. LPM performs best when applied to OLMo. What factors might contribute to this difference? For example, could OLMo’s training process or architecture make it more amenable to prototype-based moderation?
3. In Section 4.5, the important layers appear to be more distributed in OLMo, whereas they are more concentrated in other models. Could this distribution explain OLMo’s superior performance, and what might it reveal about the model’s internal representation of safety signals?
4. Why was Gaussian Discriminant Analysis (GDA) chosen as the classification method over other alternatives? Have the authors compared GDA with simpler or more expressive classifiers (e.g., logistic regression or MLPs)?

> **References**
>
> [2] *From Hard Refusals to Safe Completions: Toward Output-Centric Safety Training.*

---

> ### Author Response · Authors · 2025-11-19
> **Response (1/2)**
>
> We appreciate the reviewer’s feedback and address the mentioned points below.
>
> > My primary concern is that the proposed LPM reuses several techniques that have previously been applied to LLM trustworthiness and safety-related tasks (e.g., [1]). The main innovation appears to be aggregating classification results across multiple layers, using a method akin to LASSO for feature selection. While the empirical results are solid, the conceptual contribution seems incremental, placing the paper on the borderline of acceptance.
>
> To our knowledge, LPM is the first method to use learned prototypes for input safety assessment, which we show yields consistent, significant gains over other classification approaches as we demonstrate in. Table 1 and Appendices F and G.
>
> Moreover, we also leverage multi-layer representations for safety assessment, which we view as a non-trivial contribution. Again, as far as we know, LPM is the first application of multi-layer sparse feature selection to input moderation.
> While our method is simple, we do not view it as incremental, and the simplicity of LPM is a practical strength that makes it easy to adopt. We offer extensive evaluations and ablations of both our approach and prior methods, which constitute additional contributions of our work. LPM consistently outperforms alternative existing techniques **(see the updated Table 1, where LPM outperforms approaches such as Abdelnabi et al., which are similar to the one mentioned by the reviewer)**. Taken together, these points highlight how LPM represents a meaningful, practically relevant, and non-incremental contribution.
> > Given the simplicity of the proposed method, the paper could be strengthened by providing deeper insights into the underlying mechanisms that make LPM effective—particularly regarding what properties of the LLM’s hidden states enable accurate moderation.
>
> In our work, we have performed numerous ablations aimed at understanding why these latent representations enable such moderation (sections 4.4- discussing base vs instruct models, 4.5- analysis of most important layers for this task, 4.7- in-depth analysis of performance in low-data regimes, appendices B, H, and others). We consider these experiments to be robust empirical analyses that offer meaningful insights into the workings of our method.
>
> Moreover, our findings align with prior works that demonstrate the advantages of using multi-layer representations in neural networks [1,2]. Since our emphasis was on building an effective moderation mechanism, we focused on exploiting these representations and obtaining strong empirical results. We are happy to further discuss or expand on the internal representation studies if the reviewer is interested, and provide additional analyses on specific topics that are of interest to the reviewer.
>
> > How does LPM handle dual-use scenarios [2], where user queries may have both benign and harmful interpretations (e.g., asking for the minimum energy to ignite a firework display)?
>
> Similar to existing pure input-moderation approaches, LPM cannot distinguish between prompts with several nested intents and would flag inputs containing both harmful and harmless requests as unsafe, so we advise calibrating the refusal threshold to meet the desired performance needs in end-to-end applications. We intend our method to be a lightweight component of a whole Guardrail system, which can dynamically turn on and off the output moderation, ensuring unchanged output quality for benign prompts and saving computation. We show in Table 2 how LPM can be combined with steering methods to ensure that outputs are modified only for examples flagged as unsafe.
> Following the Reviewer’s comment, we have updated the manuscript to better highlight this nuance.
>
> > LPM performs best when applied to OLMo. What factors might contribute to this difference? For example, could OLMo’s training process or architecture make it more amenable to prototype-based moderation?
>
> We believe that better performance of OLMo representations is a result of its training. The training set of OLMo includes similar adversarial samples, thereby making its latent representations more suitable for this task. However, we note that such adversarial pretraining is not a strong requirement for the effectiveness of our method, as LPM applied to the embeddings of other models (Llama, Mistral, Qwen) also performs well (see Table 1 in our main paper).

---

> ### Author Response · Authors · 2025-11-19
> **Response (2/2)**
>
> > In Section 4.5, the important layers appear to be more distributed in OLMo, whereas they are more concentrated in other models. Could this distribution explain OLMo’s superior performance, and what might it reveal about the model’s internal representation of safety signals?
>
> This observation is valid, but we do not believe that layer distribution alone explains OLMo’s performance advantage over other models. In Figure 7b in the appendix, we show Qwen’s layer importance, which exhibits a similarly distributed pattern, indicating that we cannot draw firm conclusions about the correlation between layer distributions and model performance.
>
> A more likely explanation is that OLMo benefits from safety-oriented data in its training corpus, as discussed in our response above.
>
> > Why was Gaussian Discriminant Analysis (GDA) chosen as the classification method over other alternatives? Have the authors compared GDA with simpler or more expressive classifiers (e.g., logistic regression or MLPs)?
>
> We use GDA because it is a simple and practical method known for its good generalization [3]. We highlight that the comparisons between GDA, Logistic Regression, and Random Forest are already partially present for the OLMo2-7B model in our main work (Tables 1 and 4), as Ayub & Majumdar utilize Random Forest and Abdelnabi et al. utilize Logistic Regression.
>
> For the Reviewer’s sake, to further justify our use of GDA, we have aggregated the average safety assessment F1 scores for GDA, MLP, linear regression, random forest, and XGBoost applied to the embeddings of several models in the table below:
>
> | **Method** | **Llama-8B-Inst** | **Mistral-7B-Inst** | **OLMo2-7B-Inst** | **Qwen3-8B-Inst** |
> |-|-|-|-|-|
> | GDA | **86.25** | 84.18 | **89.44** | **85.36** |
> | Logistic Regression | 84.51 | **84.32** | 86.82 | 84.36 |
> | MLP | 85.44 | 83.93 | 87.24 | 84.02 |
> | Random Forest | 84.31 | 82.09 | 87.63 | 82.69 |
> | XGBoost | 86.02 | 82.15 | 89.37 | 84.47 |
>
> **These results highlight GDA’s strong generalization capabilities and its advantage over other supervised methods.** We have added these additional results to the updated paper in Appendix G to provide a complete picture.
>
> >The evaluation primarily focuses on input classification metrics such as F1 score. However, input moderation alone may not capture the nuanced trade-offs in real-world safety scenarios. For instance, OpenAI’s safe completions approach [2] demonstrates that maintaining safety while providing helpful responses (e.g., offering emotional support in self-harm cases) may require output-centric safety mechanisms rather than pure input filtering. Hence, the evaluation could be more convincing if it discussed the broader context and limitations of input-only moderation.
>
> We focus on input moderation, but we believe that in real-world scenarios, it should be used in conjunction with other safety guardrails, such as output-centric safety mechanisms. We are aware that input moderation is a limitation of our method. However, as shown in Table 2, the ease of utilization of LPM, together with other methods, and its input moderation performance highlight its usability as part of a larger safety system.
>
> We hope our responses address the reviewer’s concerns and make them reconsider a positive evaluation of our work. If the reviewer has any further concerns or additional points to raise, we are eager to address them.
>
> **References**
>
> [1] Masarczyk et al. “The tunnel effect: Building data representations in deep neural networks.” NeurIPS 2023
>
> [2] Szatkowski et al. “Improving continual learning performance and efficiency with auxiliary classifiers.” ICML 2025
>
> [3] Goswami et al. "Fecam: Exploiting the heterogeneity of class distributions in exemplar-free continual learning." NeurIPS 2023.

---

### Official Review · Reviewer_JQoT · 2025-10-31

**Soundness:** 3
**Presentation:** 3
**Contribution:** 3
**Rating:** 6
**Confidence:** 3

**Summary:**

This paper introduces Latent Prototype Moderator (LPM), a lightweight input-moderation method for LLMs. The approach computes Mahalanobis distances between prompt representations and safe/unsafe prototypes across multiple transformer layers, then aggregates layer-level scores with sparse weights. LPM is model-agnostic, requires no finetuning, and adds negligible inference cost. Experiments show state-of-the-art moderation performance on jailbreak and safety benchmarks, outperforming prior latent-based approaches and matching or exceeding dedicated guard models.

**Strengths:**

- Simple and principled method that leverages latent space geometry without requiring model finetuning or architectural changes. This makes the approach scalable and general. Empirically, the technique works across model sizes, architectures (including MoE) - unlike using fixed, off-the-shelf guard models.

- Thorough empirical evaluation. The authors cover multiple safety benchmarks and jailbreak settings, and compare both against guard models and prior latent-based detectors.

- Careful ablations. The analysis on number of layers selected, distance metrics (Euclidean vs Mahalanobis), data efficiency and, pre-trained vs instruction-finetuned models provide clarity on where the gains come from.

**Weaknesses:**

- Relies on labeled safety data to build prototypes. While light-weight, the pipeline still needs curated data, and the paper does not deeply examine robustness to dataset bias or domain shift.
Minor typos:
- Equations (1) and (3) must have a class index for the covariance matrices I believe (as per GDA).

**Questions:**

1) Slightly more context on the existing baselines and their approach to latent-based input moderation would benefit the reader and help positioning of the work. Is there any insight on why the LPM approach outperforms "Ayub & Majumdar (2024)", "Abdelnabi et al. (2025)" ?

---

> ### Author Response · Authors · 2025-11-19
>
> We appreciate the reviewer’s feedback and address the mentioned points below.
>
> > Relies on labeled safety data to build prototypes. While light-weight, the pipeline still needs curated data.
>
> To the best of our knowledge, the dependency on labeled data is a property of all existing moderation mechanisms. While our method is no different in this regard, we argue that it is remarkably data-efficient compared to alternatives, especially Guard models, which are trained on huge datasets.
>
> In Figure 5b, we demonstrate that **our method performs reasonably well even when used with limited data** (e.g., 100 examples per class). This allows users to create a reasonable defense mechanism at a low cost, without requiring high computational resources or large amounts of curated data, as is often the case with Guard-based solutions.
>
> > the paper does not deeply examine robustness to dataset bias or domain shift.
>
> In Table 1, we show how LPM trained on WildGuardMix performs when applied to 7 other datasets. Our method achieves very good performance even when applied to other datasets outside of the “training” distribution of WildGuardMix, which highlights the robustness of our approach to domain shifts and its generalization capabilities.
>
> Additionally, we propose how one can adapt our method to data from different distributions in a continual manner in L401-411, and we show in Figure 5a that such an approach indeed increases the overall performance in cases where we have to work in scenarios where our safety threads change and require an update to the moderation strategy.
>
> We believe these sufficiently prove the robustness of our method to dataset and domain changes, but we are also open to providing additional results if the reviewer has any other specific things in mind
>
> > Equations (1) and (3) must have a class index for the covariance matrices I believe (as per GDA).
>
> We thank the reviewer for bringing this issue to our attention. We have corrected it in the revised manuscript.
>
> > Slightly more context on the existing baselines and their approach to latent-based input moderation would benefit the reader and help positioning of the work. Is there any insight on why the LPM approach outperforms "Ayub & Majumdar (2024)", "Abdelnabi et al. (2025)" ?
>
> LPM differs from the mentioned approaches mainly by the two components: the Gaussian Discriminative Analysis (GDA) classifier and multi-level representations.
>
> The GDA classifier, in contrast to the linear regression used by the referenced works, is more robust and generalizes better [1]. It also shows favourable properties with low-resource data, as shown in the updated Appendix F, which includes a comparison between LPM, simple GDA, and Logistic Regression in terms of data scaling.
>
> Our method also leverages multi-level representation, which was motivated by previous research in other domains [2,3]. We also empirically observed that safety information is present in different layers, as shown in Figure 6. Moreover, different layers can also encode different safety threats, as presented in Figure 6a. Therefore, it is natural that using these multi-layer representations leads to improved performance over naive embedding methods.
>
> In the Table below, we aggregate results from Table 1 and Table 4 and add results for other ~8B models, showing that **GDA leads to better classification scores than standard linear regression**, which serves as the basis for these embedding methods. In Figure 5b, we also demonstrate that the model performs well in low-data regimes, which was one of our motivations for designing the model. We also demonstrate that using multi-layer representations further improves the performance.
>
> | **Method** | **Llama-8B-Inst** | **Mistral-7B-Inst** | **OLMo2-7B-Inst** | **Qwen3-8B-Inst** |
> |-|-|-|-|-|
> | LPM-SingleLayer-Euclidean | 77.17 | 77.39 | 83.61 | 74.97 |
> | LPM-MultiLayer-Euclidean | 83.73 | 80.38 | 85.74 | 78.96 |
> | LPM-SingleLayer-Mahalanobis | 86.25 | 84.18 | 89.44 | 85.36 |
> | LPM-MultiLayer-Mahalanobis | **88.30** | **87.55** | **90.18** | **85.95** |
> | Ayub & Majumdar | 84.31 | 82.09 | 87.06 | 82.69 |
> | Abdelnabi et al. | 84.51 | 84.32 | 87.35 | 84.36 |
>
> Together, these design choices lead to improved performance over previous methods and also yield other desirable properties, such as good data scaling. We thank the reviewer for raising this point and have already updated the paper to include this discussion and experiments, which will provide a clearer understanding of the reasons behind LPM performance.
>
> We hope our responses address the reviewer’s concerns and further strengthen their favourable opinion of our work. If the reviewer has any further concerns or additional points to raise, we are eager to address them.

---

> ### Author Response · Authors · 2025-11-19
>
> **References**
>
> [1] Goswami et al. "Fecam: Exploiting the heterogeneity of class distributions in exemplar-free continual learning." NeurIPS 2023.
>
> [2] Masarczyk et al. “The tunnel effect: Building data representations in deep neural networks.” NeurIPS 2023
>
> [3] Szatkowski et al. “Improving continual learning performance and efficiency with auxiliary classifiers.” ICML 2025

---

### Official Review · Reviewer_KRQX · 2025-10-31

**Soundness:** 2
**Presentation:** 2
**Contribution:** 3
**Rating:** 4
**Confidence:** 4

**Summary:**

The paper describes a method for LLM input moderation based on using internal model activations to classify prompts as harmful or safe. During classifier training, a set of labeled prompts is run through the model of interest, last token activations are captured at all layers, these are used to form class prototypes of harmful and safe prompts, and these are used to define a binary classifier. During inference, activations are captured and run through the classifier, which uses distance from the prototypes to compute a probability that the prompt is harmful. The paper compares this approach to standard guard models, as well as embedding-based and prompting-based approaches.

**Strengths:**

The approach is novel and intuitively appealing. It scales with model quality, rather than being fixed as with guard models. Many comparisons are run, there is technical analysis of the drivers of LPM performance, and cost analysis is offered.

**Weaknesses:**

The methods and evaluations used are numerous and complex, and not always clearly presented. It's also not clear that the approach offers a meaningful improvement over existing methods. Finally, some discussion of whether its implementation complexity is preferable in practice to using an off-the-shelf guard model or other alternatives would be warranted.

In Table 1, it appears that LPM does not offer clear advantages over WildGuard and embedding-based approaches, except when applied to OLMo2-7B-Instruct, and there the improvement is marginal (I am inferring that the Instruct model is being shown based on Table 7, but unlike in that table, the labeling in Table 1 is unclear).

And that model is not shown in Table 2, which is the only place that has the critical comparison to performance of the "Base Model". As the LPM method utilizes the activations of the model being used for inference, it's reasonable to wonder whether it actually improves over the refusal behavior of the inference model itself. Again the labeling is unclear as to which versions of the model are being used. Labeling could be clearer throughout that table. It's not apparent that the data in the table supports the claim that LPM meaningfully stacks with other techniques. It doesn't seem to add much to "Zheng et al. (2024)", and while the boost to F1 relative to "Unconditioned Steering" alone is substantial, the absolute F1 value is rather poor.

It's not clear that one can make a general claim about the utility of reasoning tokens based on a null effect from a single 4-B model shown in Table 3.

**Questions:**

In Table 2, why was only the WildGuardMix set used? Why is the best-performing model from Table 1 not shown? Can you clearly describe the comparisons made in each row of the table?

---

> ### Author Response · Authors · 2025-11-19
> **Response (1/2)**
>
> We appreciate the reviewer’s feedback and address the mentioned points below.
>
> > In Table 1, it appears that LPM does not offer clear advantages over WildGuard and embedding-based approaches, except when applied to OLMo2-7B-Instruct, and there the improvement is marginal (I am inferring that the Instruct model is being shown based on Table 7, but unlike in that table, the labeling in Table 1 is unclear).
>
> In the original caption to Table 1, we mention that we used embedding-based methods with OLMO embeddings; therefore, the results for embedding-based methods should be directly compared to those of LPM with OLMo as well.
>
> For the sake of completeness, we provide direct comparisons with embedding methods applied to different model embeddings with LPM. **Our method provides clear improvements over embedding-based alternatives on all tested models.**
>
> | **Method** | **Llama-8B-Inst** | **Mistral-7B-Inst** | **OLMo2-7B-Inst** | **Qwen3-8B** |
> |-|-|-|-|-|
> | LPM | **88.30** | **87.55** | **90.18** | **85.95** |
> | Ayub & Majumdar | 84.31 | 82.09 | 87.06 | 82.69 |
> | Abdelnabi et al. | 84.51 | 84.32 | 87.35 | 84.36 |
>
> We have updated our manuscript to better communicate the advantages of our method by incorporating the above results into Table 1 in the revised manuscript.
>
> > It's not clear that one can make a general claim about the utility of reasoning tokens based on a null effect from a single 4-B model shown in Table 3.
>
> At the time of the paper preparation, Qwen3-4B was the only reasonably cheap model available in both Instruct and Reasoning variants, so we provided results for these, as we still found them interesting. Since other models don’t contain instruction versions, we conducted the same experiments comparing the last vs. think tokens for Qwen’s 30B-3AB (both instruct and thinking versions), 8B, and 14B. We show those results in the Table below.
>
> | **Model** | **1.7B** | **30B-A3B-I** | **30B-A3B-T** | **4B-I** | **4B-T** | **8B** | **14B** |
> |-|-|-|-|-|-|-|-|
> | LPM(Last) | **84.95** | 87.60 | **87.40** | 86.50 | 86.33 | 86.21 | 87.64 |
> | LPM(Think) | 83.54 | - | 86.24 | - | **86.89** | **87.09** | **88.18** |
>
> With those additional results, **our conclusions from the main article still hold.** We have added the updated results to the paper to strengthen our experiments, and we are grateful to the reviewer for the inspiration.
>
> > In Table 2, why was only the WildGuardMix set used? Why is the best-performing model from Table 1 not shown? Can you clearly describe the comparisons made in each row of the table?
>
> We utilized WildGuardMix because this dataset is challenging for common models and often causes them to respond harmfully. It was created using the method described in WildTeaming[1], which enables the creation of highly effective jailbreaks even for state-of-the-art models, such as GPT and Llama.
>
> For the sake of completeness, we updated Table 2 in the manuscript with OLMo2 model results. Even for the OLMo2 model, **our method enables a better ASR-FRR tradeoff** when used as a generation-stopping mechanism or as a conditioner for output moderation methods.
>
> > Finally, some discussion of whether its implementation complexity is preferable in practice to using an off-the-shelf guard model or other alternatives would be warranted.
>
> We have provided extensive analysis of computational and memory complexity when compared to off-the-shelf models in the Discussion section, precisely in “Inference overhead.” and “Memory overhead.” (L446-L463), where we show theoretically the low complexity in both memory and computations of LPM. As discussed in the paper, LPM’s overhead on inference time computation is negligible **(< 0.001% of prompt prefill FLOPs)**, while its additional memory requirements are also a tiny fraction of the cost associated with storing the whole model **(a single-layer LPM parameters in half precision occupy around 16MB, which is negligible in comparison to billions of model weights)**. This is because, in the worst-case scenario, it only requires storing the covariance matrix and means for each layer. These are clearly cheaper, both compute-wise and memory-wise, compared to running a separate, usually large, guard model.

---

> ### Author Response · Authors · 2025-11-19
> **Response (2/2)**
>
> > It doesn't seem to add much to "Zheng et al. (2024)", and while the boost to F1 relative to "Unconditioned Steering" alone is substantial, the absolute F1 value is rather poor.
>
> Steering influences how the model responds to prompts. In our case, we utilize it to enhance answer safety, though this can lead to an increase in the False Refusal Rate (FRR). By applying conditioning, we determine whether steering should be active for a specific input. Consequently, conditioning acts primarily to lower the FRR. If the steered model produces a harmful response even when the conditioning identifies the input as unsafe, the final output will remain harmful.
>
> Conversely, when the steered model would incorrectly refuse a benign query, conditioning disables the steering because the input is flagged as safe. Therefore, overall F1 performance relies heavily on the effectiveness of the steering method itself. We highlight that our method consistently lowers the FRR, demonstrating that our approach complements output moderation techniques by improving the trade-off between ASR and FRR. While LPM's absolute F1 gain over Zheng et al. might appear small, in safety-critical applications, the reduction in False Refusal Rate (FRR) is the primary bottleneck for adoption. As shown in Table 2, **LPM significantly improves the ASR-FRR trade-off**, allowing safety steering to be deployed without destroying user experience on benign prompts.
>
> We hope our responses address the reviewer’s concerns and make them reconsider a positive evaluation of our work. If the reviewer has any further concerns or additional points to raise, we are eager to address them.
>
> ** References **
>
> [1] Jiang, Liwei, et al. "Wildteaming at scale: From in-the-wild jailbreaks to (adversarially) safer language models." NeurIPS 2024.

---

### Author Response · Authors · 2025-11-19
**Joint response**

We thank the reviewers for their insightful feedback and for recognizing the **efficiency**, **strong empirical performance**, and **intuitive design** of the Latent Prototype Moderator (LPM).

We have uploaded a revised manuscript that addresses the reviewers' questions through new experiments, expanded baselines, and clearer theoretical justifications. Below is a summary of the key updates and clarifications.

* **Our method outperforms the embedding-based baselines and guard models.** Reviewers KRQX, JQoT, 5bZL, 86ZE requested the comparison with existing embedding-based methods (e.g., Ayub & Majumdar, Abdelnabi et al.) and state-of-the-art guard models. In response, **we have expanded Table 1 to provide a direct comparison against embedding-based baselines across multiple model families** (Llama-3, Mistral, Qwen, OLMo), and **added Granite Guardian and ShieldGemma** to our evaluation. **LPM consistently outperforms the tested alternatives.**
* We have added extensive ablations (Appendices F, G, H) to empirically justify the architectural choices of LPM, addressing concerns about novelty and design. We compared GDA against Logistic Regression, MLPs, Random Forests, and XGBoost, showing that **GDA offers superior generalization and data efficiency** (Appendix G).
* We demonstrate that safety signals are distributed across layers. **Our multi-layer sparse feature selection captures these signals more effectively than single-layer approaches**, validating that LPM is a non-trivial improvement over previous works (Table 1 decomposition).
* Addressing Reviewer 86ZE, we added a comparison between "last token" and "mean pooling" representations. **The last token representation consistently yields higher performance**, as it encodes the specific intent of the subsequent generation in instruction-tuned models (Appendix H).
* We have addressed the Reviewer KRQX’s concern regarding the **generalizability of our findings on reasoning models**, expanding our analysis with **Qwen 30B-A3B (both Instruct and Thinking variants), 8B, and 14B**. With those additional results, **our conclusions from the main article remain unchanged.**
* We clarified the computational advantages of LPM (as requested by reviewer KRQX) in the Discussion. LPM introduces negligible overhead (**\<0.001% FLOPs**) and requires minimal memory (\~16MB per layer), making it orders of magnitude more efficient than deploying separate guard models (such as LlamaGuard or WildGuard).
* We addressed reviewer JQoT's concerns about data reliance by demonstrating that **LPM performs robustly with as few as 100 labeled examples** (Figure 5b). Furthermore, we validated robustness to domain shift by training on WildGuardMix and evaluating on 7 diverse datasets, where LPM maintained high performance.

We believe these revisions firmly establish LPM as a novel, efficient, and state-of-the-art solution for input moderation.

---

> ### Author Response · Authors · 2025-11-27
> **Discussion period follow-up**
>
> We are grateful for your constructive feedback, which has allowed us to strengthen our work with additional experiments and expanded insights. As the discussion period draws to a close, please let us know if you have any remaining concerns. We would be happy to engage in further discussion to address them. If you feel that we have adequately addressed your concerns, we would be grateful if you would increase your score.

---

### Note · Program_Chairs · 2026-01-17
**Submission Desk Rejected by Program Chairs**

The following references in this submission do not refer to real documents and/or have major errors in bibliographic information:

 Nicholas Carlini and et al. Aligned but unsafe: Investigation of jailbreaks in ai systems. arXiv preprint arXiv:2305.18526, 2023.
Ananya Kamath, Shiyu Sun, Xiaoyang Gu, Rui Zhang, Yixuan Liu, Matthias Hein, and Yixuan Li. Openood: Benchmarking generalized out-of-distribution detection. arXiv preprint arXiv:2302.12229, 2023.
Jason Wei and et al. Jailbreaking large language models via prompt engineering. arXiv preprint arXiv:2303.16200, 2023.